# Comparative analysis of salivary antimicrobial resistance genes in dental students: A PCR and questionnaire study

**Marwan Mansoor Mohammed**[1,2]*, **Priyadharshini Sekar**[2], **Jahida Al Jamal**[1], **Lujayn Abu Taha**[1], **Asma Bachir**[2], **Sausan Al Kawas**[1,2]

**1** Department of Oral and Craniofacial Health Sciences, College of Dental Medicine, University of Sharjah, Sharjah, United Arab Emirates, **2** Research Institute for Medical and Health Sciences, University of Sharjah, Sharjah, United Arab Emirates

* mmmohammed@sharjah.ac.ae

**Data Availability Statement:** All relevant data are available within the paper. Additional data are also

## Abstract

### Introduction

Antimicrobial resistance (AMR) is a major global healthcare challenge, with limited treatment options due to the decline in new antibiotics. The human oral cavity, home to diverse bacteria, is crucial for maintaining oral and systemic health. Recent studies suggest that saliva may serve as a reservoir for AMR genes. However, there is a lack of research on this topic in the UAE and most Middle Eastern countries. This study investigated the presence of AMR genes in saliva from forty 5th-year dental students and forty 1st-year dental students.

### Materials & methods

Demographic and health information was collected via a 28-question structured questionnaire. Real-Time PCR was used to detect a panel of preselected AMR genes in bacterial DNA from saliva samples.

### Results

Participants' ages ranged from 20 to 31 years, with 41 females and 39 males. The prevalence of AMR genes varied: blaCTX-M grp 1 (29%), blaCTX-M grp 9 (85%), blaCTX-M grp 8 (39%), blaOXA-48 (69%), blaKPC-1 (6%), blaVIM (49%), DHA (53%), ACC (25%), MOX (59%), armA (83%), and rmtB (63%). There were no significant differences in AMR gene prevalence between 5th-year and 1st-year students or between male and female students.

### Conclusion

The study revealed a high occurrence of AMR genes in the oral microbiome. Comprehensive metagenomic analysis is recommended to further evaluate the prevalence and relative abundance of these genes in the UAE population. Establishing a database for these ARGs could aid in effective future monitoring.

available on the Open Science Framework (OSF) at https://osf.io/34jhd/.

**Funding:** "This research was supported by the Research Funding Department, University of Sharjah, Sharjah, UAE. (Grant No. 2101100144)".

**Competing interests:** The authors have declared that no competing interests exist.

## Introduction

Antibiotic resistance (AMR) is increasingly becoming a major global healthcare challenge. With the decrease in the number of new antibiotics introduced over the past three decades, few treatment options remain for clinicians [1]. According to the CDC's Antibiotic Resistance Threats in the United States, 2019, more than 2.8 million antibiotic-resistant infections occur annually in the U.S., resulting in over 35,000 deaths [2].

Several bacteria that construct a complex microbial ecosystem with competition and symbiosis are present in the human oral cavity [3]. The human oral cavity hosts a heterogeneous microbiota that includes commensal bacteria essential for the preservation of both oral and systemic health [4]. It is one of the most complex microbial ecosystems in the human body, with more than 700 bacterial species identified [5]. This complex microbial ecosystem, characterized by competition and symbiosis, creates suitable conditions for horizontal gene transfer [6].

One factor that contributes to antimicrobial resistance is the use of antimicrobial agents, regardless of whether they are appropriate or not. This usage increases the selective pressure on bacteria to develop resistance. Therefore, the effects of proper regulations on antibiotic use in each country should be observable in the oral resistome of individuals from that country, especially if higher levels of exposure lead to stronger selection. This evidence should demonstrate a direct connection between controlling access to clinically relevant antibiotics and the resistome in the human oral cavity [7].

Several recent studies have been done in the last few years to detect the ARGs in the saliva of the participants and the results showed that these genes are widely spread and the oral cavity work as a reservoir for these genes [7–9]. While AMR in clinical settings has received significant attention, recent studies have begun to explore its prevalence and implications in non-clinical settings, such as the oral cavities of dentists and dental students. Research suggests that due to their frequent contact with diverse microorganisms in clinical environments, dentists and dental students may harbor antimicrobial-resistant bacteria in their saliva [10, 11]. This phenomenon is concerning because saliva serves as a reservoir for microorganisms, including potentially pathogenic strains. The presence of antimicrobial-resistant bacteria in saliva increases the risk of transmission to patients during dental procedures, posing potential risks to patient safety and treatment efficacy [12]. Numerous factors contribute to the onset and dissemination of antimicrobial resistance within the saliva of dental students. These encompass the overutilization and inappropriate application of antibiotics, insufficient adherence to infection control protocols, exposure to antimicrobial agents inherent in dental materials, and proximity to patients harboring or affected by resistant microbial strains. Furthermore, the dispersion of saliva via aerosols generated during dental interventions may serve as a mechanism for the inter-individual transmission of antimicrobial-resistant bacteria.

Enzyme-mediated antibiotic resistance represents a pivotal mechanism utilized by bacteria to counteract the efficacy of antimicrobial agents. A prominent illustration is the production of β-lactamases, enzymes capable of hydrolyzing the β-lactam ring present in numerous antibiotics, encompassing penicillin, cephalosporins, and carbapenems. These enzymes are categorized into diverse classes based on their structural and functional attributes, including extended-spectrum β-lactamases (ESBLs), AmpC β-lactamases, and carbapenemases, each exhibiting distinct substrate specificities [13]. Through β-lactam ring cleavage, β-lactamases undermine the antimicrobial activity of antibiotics, fostering treatment failure and the persistence of infections [14]. Furthermore, bacteria may employ alternative enzymatic mechanisms to confer antibiotic resistance, such as target modification or antibiotic inactivation via chemical alterations or by non-enzymatic mechanisms such as efflux pumps [14]. 16S rRNA

methyltransferase that catalyzes the methylation of the 30S ribosomal subunit, thereby obstructing aminoglycoside binding to the bacterial ribosome. This modification reduces the affinity of aminoglycosides for their target site, diminishing their bactericidal activity [15]. The emergence and dissemination of drug resistant bacterial strains have been facilitated by various factors, including the overuse and misuse of antibiotics, particularly broad-spectrum cephalosporins and fluoroquinolones, which exert selective pressure favoring the survival and proliferation of resistant bacteria. Additionally, mobile genetic elements, such as plasmids and transposons, play a pivotal role in the horizontal transfer of β-lactamases and 16S rRNA methyltransferases genes between different bacterial species, further contributing to the spread of resistance [16]. In this study, we selected specific AMR genes based on their clinical relevance, known prevalence in the oral microbiome, and potential impact on infection control in dental settings [7]. The genes analyzed—such as blaCTX-M (ESBL genes), carbapenemases (blaOXA-48, blaKPC), AmpC-type beta-lactamases (DHA, MOX), and aminoglycoside resistance genes (armA, rmtB)—are associated with resistance to widely used antibiotics, including those important for treating both oral and systemic infections. These genes were chosen due to their potential for horizontal gene transfer within the oral biofilm, and their importance in AMR surveillance in the UAE, where dental students may serve as a unique population exposed to resistant strains in clinical environments. Our analysis provides foundational insights relevant to infection control and AMR monitoring in regional healthcare settings [9, 17, 18].

In this study, we conducted a comparative analysis of salivary antimicrobial resistance in first and fifth-year dental students, using a combination of questionnaire surveys and real-time PCR techniques. The research aimed to elucidate potential differences in antimicrobial resistance profiles between these two cohorts, considering their varying levels of exposure to clinical environments and antimicrobial agents. Questionnaire surveys assessed factors such as antibiotic usage, infection control practices, and exposure to antimicrobial agents in dental materials. Salivary samples were collected from participants, and real-time PCR analysis was employed to detect and quantify antimicrobial resistance genes. Our findings provide valuable insights into the prevalence and determinants of antimicrobial resistance in dental students' saliva, highlighting potential areas for targeted intervention and education to mitigate the spread of antimicrobial resistance in dental settings. This study aligns with the UAE National Action Plan on Antimicrobial Resistance, implemented in 2017.

## Materials and methods

### Study population

Saliva samples were collected from forty $5^{th}$ year dental students, attending the University Dental Hospital Sharjah (UDHS), and from the same number of 1st year students who are not attending the clinics yet. The participant recruitment was done form $1^{st}$ of August 2022 to the end of December 2022. The balance between the males and females was considered.

The exclusion criteria for the study included any antibiotic therapy within the last three months and any systemic disease that could influence the composition of oral bacteria, such as diabetes mellitus.

The study was reviewed and approved by the University of Sharjah Research Ethics Committee (approval number: REC-21-09-16-01) and was conducted in accordance with the ethical standards outlined in the Declaration of Helsinki (1964) and its subsequent amendments. Written informed consent was obtained from all participants prior to their involvement in the study. All participants were adults and there was no obtained consent from parents or guardians.

## Questionnaire

A self-administered questionnaire consisting of 28 questions was the tool used in this study. The questionnaire was accompanied by a participant information sheet to clarify the objective of the study, assure anonymous and voluntary participation, and confirm that the responses will be confidential, and accessible only by the authors. The questionnaire was adopted from previously used survey [11] and modified by the authors and pilot tested on a group of ten dental students for validation. Afterwards, adjustments were made to ensure a clear and comprehensive version of the questionnaire.

The study participants completed a comprehensive questionnaire to gather information about their general health status, medication usage, and personal habits. The questionnaire explored their general health, use of antibiotics and other medications in the recent past, as well as tobacco consumption habits. It delved into their dental health history, oral hygiene practices and routines, and commitment to maintaining personal hygiene standards. The questionnaire aimed to provide a comprehensive assessment of the participants' overall health, oral hygiene practices, and their knowledge and attitudes toward infection control measures within the dental practice environment. This information would assist in identifying potential areas for improvement and developing strategies to enhance infection control practices among dental professionals. The questionnaire was prepared and filled on paper by the participants during the saliva samples collection session.

## Saliva sample collection

Each study participant donated a whole unstimulated saliva specimen. The participants were informed about the procedure of collecting saliva beforehand in a separate session, and the information was also stressed prior to the sample collection. The subjects were informed not to eat, drink, or use any form of nicotine within the last hour before sampling. Then the participants spat saliva frequently into a sterile collecting tube for five minutes (2–3 ml). Saliva samples were stored in -80˚C freezer for further analysis.

## DNA extraction from saliva samples

The DNA extraction from the saliva samples was performed using a commercial DNA extraction kit (NucleoMag, DNA Microbiome, Macherey-Nagel, Germany) as per the instructions provided in the kit. Briefly, 200 mg of saliva sample was transferred to a 2 mL microcentrifuge tube and was heated at 70˚C for 5 min, followed by vigorous vortexing for 10 minutes. This was then incubated for 5 min at room temperature, centrifuged for 5 min at 11000 X $g$ and the supernatant was transferred to a fresh tube. 150 μL of Mic was added to the supernatant, vortexed for 5 seconds, incubated at 4˚C for 10 min, centrifuged for 5 min at 11000 X $g$. The supernatant was transferred to a fresh tube. To about 500 μL of lysate, 25 μL NucleoMag® B-Beads was added followed by 310 μL MI2. This was mixed by shaking for 5 min at RT. The tube was then placed in a magnetic separation stand for 5 min and the supernatant was removed. Then 600 μL MI3 was added, resuspend by shaking for 2 min at RT, placed on the magnetic separation stand for 2 min and the supernatant was removed. This was followed by addition of 600 μL MI3 to the tube, resuspension by shaking for 2 min at RT, then separation by placing on the magnetic stand for 2 min and then the supernatant was removed. The process was repeated with 600 μL MI4, followed by 600 μL with 70% EtOH. The beads were then air dried for 10–15 min at RT while being placed on the magnetic separation stand. Then, elution was done by adding 50-100 μL MI5, shaking for 5 min at RT, followed by magnetic separation for 2 min, and the supernatant containing the DNA was transferred to a fresh tube. The

purity and concentration of the eluted DNA were measured using a Nano-Drop™ spectrophotometer and stored at -20˚C until used further.

## Real-time quantitative polymerase chain reaction

The genes contributing to antibiotic resistance by ESBL—$bla_{CTX-M1}$, $bla_{CTX-M9}$, $bla_{CTX-M8}$, carbapenemase–$bla_{OXA-48}$, $bla_{KPC}$, $bla_{VIM}$ and AmpC beta-lactamase–DHA, ACC, MOX and genes were aminoglycoside resistance–$armA$ and $rmtB$ were quantified in the DNA sample obtained from saliva of the dental students. Real-Time qPCR amplification was performed with a SYBR green master mix - 5X Hot FirePol EvaGreen qRT-PCR SuperMix (Solis Biodyne, Tartu, Estonia) in QuantStudio 3 Real-Time PCR System (Applied Biosystems, CA, USA). The primers for the study are mentioned in Table 1. Real Time PCR was done with standard curve, with the following cycling conditions: 50˚C for 2 min, 95˚C for 10 min, 40 cycles of 95˚C for 15 seconds, 60˚C for 1 minute, followed by melt curve stage—95˚C for 15 seconds, 60˚C for 1 minute, 95˚C for 1 second. Ct value of each PCR reaction was recorded.

## Data analysis

The software IBM SPSS Statistics (Version 28) was used to analyze the questionnaire answers. Statistical analysis with descriptive statistics using frequencies was performed. Real-Time qPCR data was analyzed using GraphPad Prism 5. The Ct values were plotted as mean ± standard deviation for the two study groups. One-way ANOVA with Tukey's multiple comparison test, unpaired t test or Chi-square test were used where necessary to compare the resistance genes.

**Table 1. List of primers used for real-time qPCR in the study.**

| Family | Gene | Primer | References |
|---|---|---|---|
| **Extended-spectrum β-lactamases (ESBLs)** | $bla_{CTX-M}$ *grp 1* | F: AAAAATCACTGCGCCAGTTC | [19] |
| | | R: AGCTTATTCATCGCCACGTT | |
| | $bla_{CTX-M}$ *grp 9* | F: CAAAGAGAGTGCAACGGA TG | [19] |
| | | R: ATTGGAAAGCGTTCATCACC | |
| | $bla_{CTX-M}$ *grp 8* | F: TCGCGTTAAGCGGATGAT GC | [19] |
| | | R: AACCCACGATGTGGGTAC | |
| **Carbapenemase** | $bla_{OXA-48}$ | F: GCGTGGTTAAGGATGAACAC | [20] |
| | | R CATCAAGTTCAACCCAACCG | |
| | $bla_{KPC-1}$ | F: CGTCTAGTTCTGCTGTCTTG | [20] |
| | | R: CTTGTCATCCTTGTTAGGCG | |
| | $bla_{VIM}$ | F: GATGGTGTTTGGTCGCATA | [20] |
| | | R: CGAATGCGCAGCACCAG | |
| **AmpC beta-lactamase** | *DHA-1, DHA-2* | F: AACTTTCACAGGTGTGCTGGGT | [17] |
| | | R: CCGTACGCATACTGGCTTTGC | |
| | *ACC* | F: AACAGCCTCAGCAGCCGGTTA | [17] |
| | | R: TTCGCCGCAATCATCCCTAGC | |
| | *MOX-1, MOX-2, CMY-1, CMY-8 to CMY-11* | F: GCTGCTCAAGGAGCACAGGAT | [17] |
| | | R: CACATTGACATAGGTGTGGTGC | |
| **Aminoglycoside resistance** | *armA* | F: ATTCTGCCTATCCTAATTGG | [18] |
| | | R: ACCTATACTTTATCGTCGTC | |
| | *rmtB* | F: GCTTTCTGCGGGCGATGTAA | [18] |
| | | R: ATGCAATGCCGCGCTCGTAT | |

**Table 2. Students' exposure to antibiotics in their lifetime.**

| | | Antibiotic exposure in the participant's lifetime | | | | |
|---|---|---|---|---|---|---|
| | | Never | 1–2 times | 3–10 times | > 10 times | Total |
| 1st year dental student | N | 6 | 6 | 21 | 7 | 40 |
| | % | 15.0% | 15.0% | 52.5% | 17.5% | 100.0% |
| 5th year dental student | N | 4 | 9 | 20 | 7 | 40 |
| | % | 10.0% | 22.5% | 50.0% | 17.5% | 100.0% |
| Total | N | 10 | 15 | 41 | 14 | 80 |
| | % | 12.5% | 18.8% | 51.3% | 17.5% | 100.0% |

## Results

Most of the participants 98.7% (n = 79) described their general health as good or very good. This is also applied on their dental health as 92.6% (n = 74) chose good or very good to describe their dental health status. Total of 87.5% (n = 35) answered that they do not have a chronic illness that requires regular medical treatment. The same percentage 87.5% (n = 35) from both groups of students had not taken antibiotics three months prior to the commencement of this study. Both groups of the students had a comparable exposure to antibiotics in their lifetime (Table 2). Approximately 17.5% (n = 7) of $1^{st}$ year students and 42.5% (n = 17) of $5^{th}$ year students were observed to engage in smoking behavior for two years or more. Both the groups of students practice hand washing after different activities and procedures in dental clinic.

In general, 78 (97.5%) samples were positive for at least one of the genes screened, and the different genes showed varied prevalence within the screened samples (Table 3).

### Determinants of ESBL mediated antibiotic resistance

Saliva samples were collected from 40 first year and 40 fifth year dental students. A total of 23 samples were positive for $bla_{CTX-M1}$, 68 positive for $bla_{CTX-M9}$ and 31 positive for $bla_{CTX-M8}$ (Fig 1). It was observed that among the three ESBL-producing genes targeted, $bla_{CTX-M9}$ was found to be significantly higher ($p<0.05$) prevalent in 85% of the total samples when compared to $bla_{CTX-M1}$ and $bla_{CTX-M8}$. This was followed by $bla_{CTX-M8}$ which was significantly more prevalent ($p<0.05$) in 39% compared to $bla_{CTX-M1}$ which was prevalent in 29% samples (Fig 2). Among the first-year dental students $bla_{CTX-M1}$ was present in 11 samples, $bla_{CTX-M9}$ in 33

**Table 3. The prevalence of AMR genes in the samples.**

| Gene | Prevalence |
|---|---|
| blaCTX-M1 | 29% |
| blaCTX-M9 | 85% |
| blaCTX-M8 | 39% |
| blaOXA-48 | 69% |
| blaKPC-1 | 6% |
| blaVIM | 49% |
| DHA | 53% |
| ACC | 25% |
| MOX | 59% |
| armA | 83% |
| rmtB | 63% |

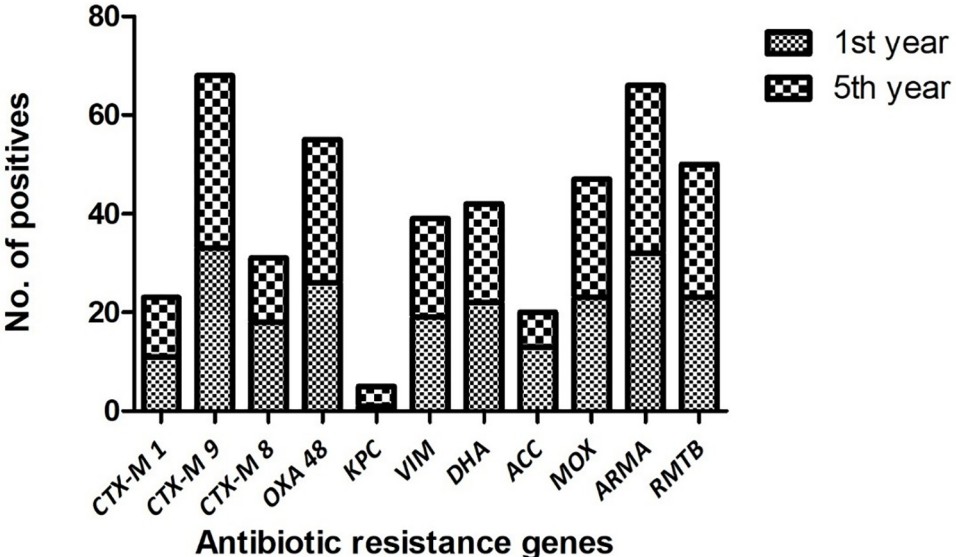

**Fig 1. Total prevalence of antibiotic resistance genes among the dental students.**

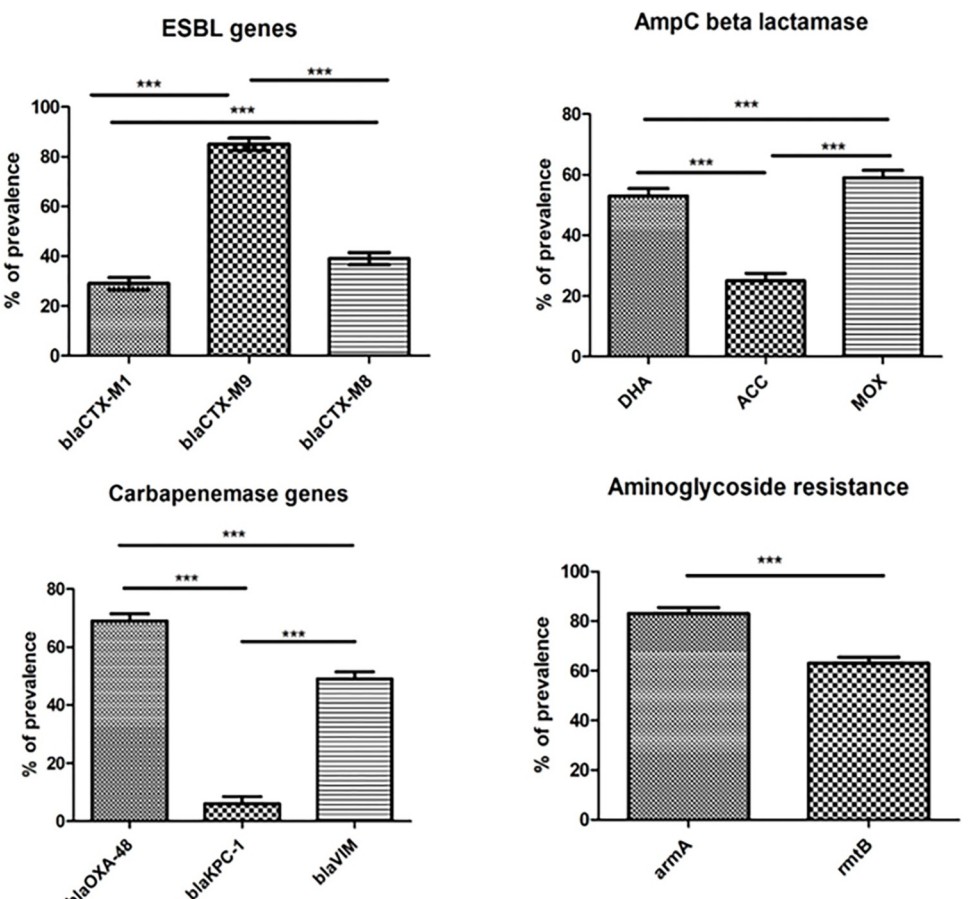

**Fig 2. Prevalence of ESBL, carbapenemase, AmpC, and aminoglycoside resistance genes among the dental students.**

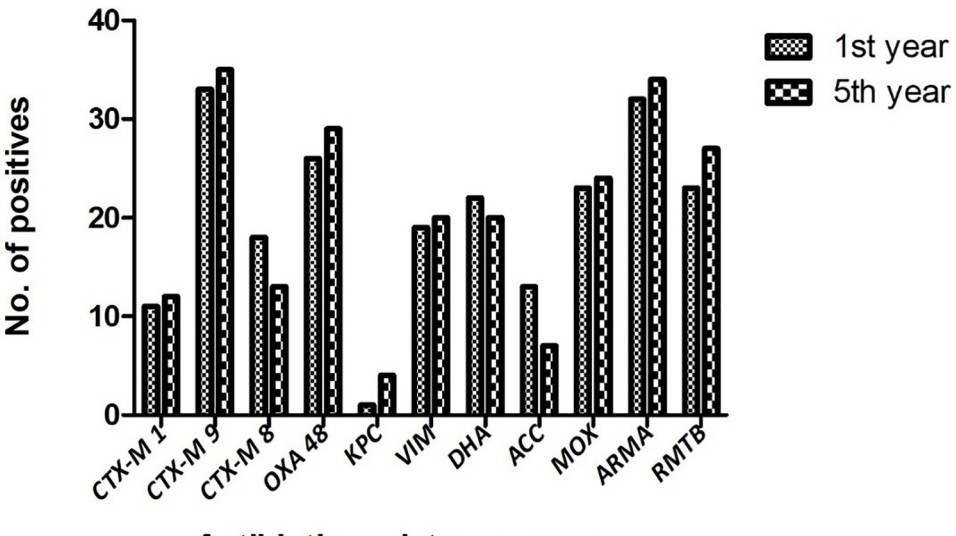

**Fig 3. Distribution of antibiotic resistance genes among first- and fifth-year dental students.**

and $bla_{CTX-M8}$ in 18 samples. Among the fifth-year students, $bla_{CTX-M1}$ was present in 12 samples, $bla_{CTX-M9}$ in 35 and $bla_{CTX-M8}$ in 13 samples (Fig 3). The mean ± SD Ct value for $bla_{CTX-M1}$ is 14.63 ± 6.16, $bla_{CTX-M9}$ is 35.68 ± 1.8 and $bla_{CTX-M8}$ is 18.78 ± 8.95 among both the groups of dental students (Fig 4). Among the first-year dental students, the mean ± SD Ct value for $bla_{CTX-M1}$ is 12.36 ± 5.47, $bla_{CTX-M9}$ is 35.57 ± 1.91and $bla_{CTX-M8}$ is 20.46 ± 9.79. Among the fifth-year dental students, the mean ± SD Ct value for $bla_{CTX-M1}$ is 17 ± 6.15, $bla_{CTX-M9}$ is 35.79 ± 1.7 and $bla_{CTX-M8}$ is 17.1 ± 7.69 (Fig 5).

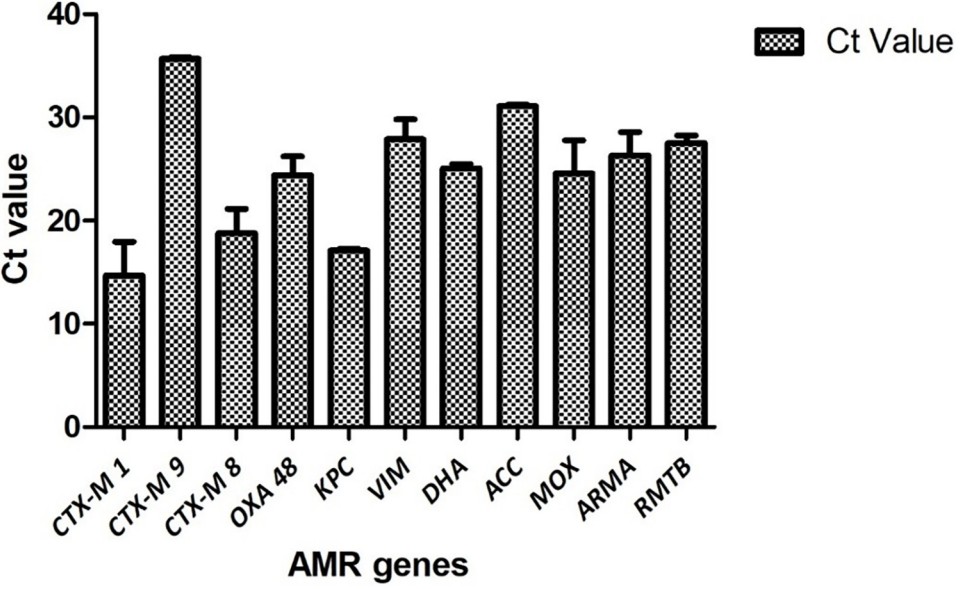

**Fig 4. Mean ± SD of Ct value of antibiotic resistance genes among the dental students.**

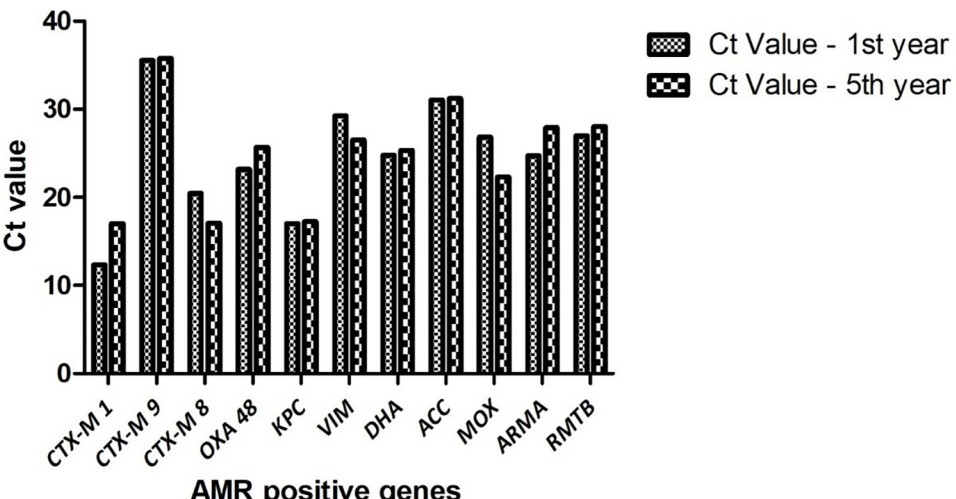

**Fig 5. Comparison of Ct value of antibiotic resistance genes among the first- and fifth-year dental students.**

## Determinants of carbapenemase-mediated antibiotic resistance

A total of 55 samples were positive for $bla_{OXA-48}$, 5 positive for $bla_{KPC}$ and 39 positive for $bla_{VIM}$ (Fig 1). It was observed that among the three carbapenemase producing genes targeted, $bla_{OXA-48}$ was found to be significantly higher (p<0.05) prevalent in 69% of the total samples when compared to $bla_{VIM}$ and $bla_{KPC-1}$. This was followed by $bla_{VIM}$ which was significantly more prevalent (p<0.05) in 49% compared to $bla_{KPC-1}$ which was prevalent in 6% samples (Fig 2). Among the first-year dental students $bla_{OXA-48}$ was present in 26 samples, $bla_{KPC}$ in 1 sample and $bla_{VIM}$ in 19 samples. Among the fifth-year students, $bla_{OXA-48}$ was present in 29 samples, $bla_{KPC}$ in 4 samples and $bla_{VIM}$ in 20 samples (Fig 3). The mean ± SD Ct value for $bla_{OXA-48}$ is 24.42 ± 8.55, $bla_{KPC}$ is 17.14 ± 12.06 and $bla_{VIM}$ is 27.89 ± 8.22 among both the groups of dental students (Fig 4). Among the first-year dental students, the mean ± SD Ct value for $bla_{OXA-48}$ is 23.17 ± 9.18 and $bla_{VIM}$ is 29.25 ± 7.31. Among the fifth-year dental students, the mean ± SD Ct value for $bla_{OXA-48}$ is 25.68 ± 7.94, $bla_{KPC}$ is 17.24 ± 14.77 and $bla_{VIM}$ is 26.53 ± 8.96 (Fig 5).

## Determinants of AmpC beta lactamase mediated antibiotic resistance

A total of 42 samples were positive for DHA, 20 positive for ACC and 47 positive for MOX (Fig 1). It was observed that among the three AmpC beta lactamase-producing genes targeted, MOX was found to be significantly higher (p<0.05) prevalent in 59% of the total samples when compared to DHA and ACC. This was followed by DHA which was significantly more prevalent (p<0.05) in 53% compared to ACC which was prevalent in 25% samples (Fig 2). Among the first-year dental students 22 samples were positive for DHA, 13 positive for ACC and 23 positive for MOX. Among the fifth-year students 20 samples were positive for DHA, 7 positive for ACC and 24 positive for MOX (Fig 3). The mean ± SD Ct value for DHA is 25.08 ± 9.02, ACC is 31.12 ± 6.19 and MOX is 24.57 ± 10.07 (Fig 4). Among the first-year dental students, the mean ± SD Ct value for DHA is 24.8 ± 10.1, ACC is 31.04 ± 6.44 and MOX is 26.83 ± 9.78. Among the fifth-year dental students, the mean ± SD Ct value for DHA is 25.36 ± 7.99, ACC is 31.20 ± 6.28 and MOX is 22.30 ± 10.07 (Fig 5).

## Determinants of aminoglycoside resistance

A total of 66 samples were positive for *armA* and 50 were positive for *rmtB* (Fig 1). It was observed that among the two aminoglycoside resistance genes targeted, *armA* was found to be

**Table 4. Correlations between AMR, good health, non-usage of antibiotic and non-smoking.**

|  | First year | Fifth year | Row Totals |
|---|---|---|---|
| Good health | 20 (16.84) [0.59] | 12 (15.16) [0.66] | 32 |
| Non-smokers | 31 (28.41) [0.24] | 23 (25.59) [0.26] | 54 |
| 3 month non usage of antibiotic | 35 (36.83) [0.09] | 35 (33.17) [0.10] | 70 |
| Beta Lactamase | 18 (18.94) [0.05] | 18 (17.06) [0.05] | 36 |
| Aminoglycoside | 27 (29.99) [0.30] | 30 (27.01) [0.33] | 57 |
| *Column Totals* | 131 | 118 | **249 (Grand Total)** |

The chi-square statistic is 2.6716. The p-value is .614185. The result is not significant at p < .05. The table provides the following information: the observed cell totals, (the expected cell totals) and [the chi-square statistic for each cell].

significantly higher (p<0.05) prevalent in 83% of the total samples when compared to *rmtB* which was prevalent in 63% samples (Fig 2). Among the first-year dental students 32 were positive for *armA* and 23 were positive for *rmtB*. Among the fifth-year dental students 34 were positive for *armA* and 27 were positive for *rmtB* (Fig 3). The mean ± SD Ct value for *armA* is 26.32 ± 6.77 and *rmtB* is 27.51 ± 7.92 (Fig 4). Among the first-year dental students, the mean ± SD Ct value for *armA* is 24.71 ± 5.88 and *rmtB* is 26.98 ± 10.64. Among the fifth-year dental students, the mean ± SD Ct value for *armA* is 27.92 ± 7.24 and *rmtB* is 28.04 ± 4.92 (Fig 5).

Correlations between Beta lactamase producing genes, aminoglycoside resistance gene, good health, non-usage of antibiotic and non-smokers.

The Chi-square test was conducted to examine potential associations between participant year group (first year vs. fifth year) and various characteristics, including health status, smoking habits, recent antibiotic usage, the presence of beta lactamase producing genes–ESBL, AmpC beta lactamase, Carbapenemase and aminoglycoside resistance genes. An average of the presence of all the beta lactamase producing genes and an average of all the aminoglycoside resistance genes were included for the analysis. Observed values for each characteristic were close to the expected counts, with minimal Chi-square contributions across all cells: good health (0.59 and 0.66), non-smokers (0.24 and 0.26), 3-month non-usage of antibiotics (0.09 and 0.10), Beta-lactamase gene presence (0.05 in both groups), and Aminoglycoside gene presence (0.30 and 0.33). These low contributions indicate no significant differences in the distribution of these characteristics between first- and fifth-year participants. The overall Chi-square statistic was 2.6716, with a p-value of 0.614185, indicating that the result is not statistically significant at p < 0.05. Therefore, there is no evidence of an association between participant year group and the various characteristics (Table 4).

## Discussion

There are a few studies that investigated the prevalence of antibiotic resistance genes (ARGs) in saliva [21–24]. Some of these studies investigated the AMR genes in specific bacterial strains and others did not specify the strains [7]. The high prevalence of certain ARGs, such as $bla_{CTX-M9}$ and $bla_{OXA-48}$, highlights the potential for dissemination of antimicrobial resistance within the oral microbiome. These findings align with previous studies that have reported similar trends in ARG distribution, indicating the persistence and widespread nature of certain resistance mechanisms [7].

Extended-spectrum beta-lactamases (ESBLs) of the CTX-M family, including $bla_{CTX-M1}$, $bla_{CTX-M9}$, and $bla_{CTX-M8}$ variants, represent a significant challenge in antimicrobial therapy due to their ability to confer resistance to a broad spectrum of beta-lactam antibiotics. These

enzymes, primarily produced by Gram-negative bacteria, are characterized by their enhanced hydrolytic activity against extended-spectrum cephalosporins and monobactams. The $bla_{CTX-M1}$, $bla_{CTX-M9}$, and $bla_{CTX-M8}$ genes encode beta-lactamases that belong to molecular class A, specifically the CTX-M-type beta-lactamases. These enzymes exhibit a preference for hydrolyzing cefotaxime, although they also confer resistance to other third-generation cephalosporins such as ceftazidime [14]. It is noteworthy that 68 samples tested positive for $bla_{CTX-M9}$ among the dental students. However, the mean ± SD Ct value was 35.68 ± 1.8, indicating a low initial concentration of the target gene being amplified despite the relatively higher prevalence of this CTX-M variant. In contrast, the mean ± SD Ct value for $bla_{CTX-M1}$ was 14.63 ± 6.16, with its presence observed in 11 first-year students and 12 fifth-year students. Although the prevalence of $bla_{CTX-M1}$ was comparable between the two groups, the low Ct value suggests a higher concentration of the target gene. Additionally, $bla_{CTX-M8}$ was slightly more prevalent among first-year students (18 samples) compared to fifth-year students (13 samples), with a moderate concentration observed in positive samples (mean ± SD Ct value of 18.78 ± 8.95). However, functional assays are required to confirm the enzymatic activity of the $bla_{CTX-M}$ genes to determine the extended hydrolytic activity on third generation cephalosporins.

The emergence and dissemination of carbapenemase-producing Enterobacteriaceae (CPE), notably those harboring the OXA-48, KPC, and VIM carbapenemase enzymes, pose significant challenges in clinical settings. OXA-48, a class D carbapenemase, has gained attention due to its ability to hydrolyze carbapenems and exhibit limited susceptibility to inhibition by commonly used β-lactamase inhibitors. Similarly, *Klebsiella pneumoniae* carbapenemase (KPC), a class A serine β-lactamase, confers resistance to carbapenems and other β-lactam antibiotics, posing a considerable threat to antimicrobial therapy. Additionally, Verona integron-encoded metallo-β-lactamase (VIM), a class B carbapenemase, utilizes zinc ions to hydrolyze carbapenems, rendering them ineffective against CPE [25]. It is concerning to observe that 55 dental students carried $bla_{OXA-48}$, with a similar distribution of 26 positive cases among first-year students and 29 positive cases among fifth-year students. The mean ± SD Ct value for $bla_{OXA-48}$ was 24.42 ± 8.55, suggesting a moderate prevalence of the carbapenemase enzyme. The identification of only 5 students testing positive for $bla_{KPC}$, with one case reported among first-year students and four cases among fifth-year students, indicating a low prevalence. However, it was concerning to observe that the mean ± SD Ct value for blaKPC was 17.14 ± 12.06, indicating a significant presence of this target gene among positive cases. Additionally, the presence of blaVIM was comparable between first-year and fifth-year dental students, with a mean ± SD Ct value of 26.53 ± 8.96, suggesting a low to moderate prevalence of this enzyme.

AmpC beta-lactamases, including DHA, ACC, and MOX variants, represent a substantial challenge in antimicrobial therapy due to their capacity to confer resistance to a wide range of beta-lactam antibiotics. These enzymes, predominantly produced by Gram-negative bacteria such as Enterobacter spp., Citrobacter spp., and Morganella spp., possess the ability to hydrolyze penicillins, cephalosporins, and monobactams. DHA (Dhahran Hospital variant), ACC (Ambler class C cephalosporinase), and MOX (*Morganella morganii*-derived cephalosporinase) are among the most clinically relevant AmpC beta-lactamase variants encountered in healthcare settings. The widespread prevalence of AmpC beta-lactamases is further compounded by their inherent chromosomal location and inducibility, rendering them refractory to traditional β-lactamase inhibitors and contributing to treatment failure in infections caused by multidrug-resistant bacteria [26]. The predominant determinants of AmpC beta-lactamase were found to be DHA and MOX, with 42 and 47 positive samples, respectively. The prevalence of both these determinants was nearly equivalent among first and fifth-year dental

students. The target genes for DHA and MOX exhibited a moderate concentration, with mean ± SD Ct values of 25.08 ± 9.02 and 24.57 ± 10.07, respectively. Conversely, only 20 samples tested positive for ACC, with a higher prevalence observed among first-year students, present in 13 of them compared to only 7 among fifth-year students. The ACC genes were detected in lower concentrations, with a mean ± SD Ct value of 31.12 ± 6.19.

Aminoglycoside resistance, facilitated by 16S rRNA methyltransferases, constitutes a prominent strategy employed by bacteria to counteract the effects of these antimicrobial agents. These enzymes, encoded by genes such as *armA* and *rmtB*, catalyze the methylation of specific adenine residues within the 16S ribosomal RNA, thereby impeding the binding of aminoglycosides to the bacterial ribosome. This alteration reduces the effectiveness of aminoglycosides, compromising their ability to impede protein synthesis and exert bactericidal actions. Notably, the *armA* and *rmtB* genes are frequently located on mobile genetic elements, enabling their horizontal transfer between bacterial strains and species, thereby facilitating the dissemination of aminoglycoside resistance [27]. A notable occurrence of *armA* and *rmtB* was noted among the dental students, with 66 and 50 samples testing positive, respectively. Their distribution exhibited similarity, with both genes being detected in a moderately low concentration, as indicated by the mean ± SD Ct value for *armA* of 26.32 ± 6.77 and for *rmtB* of 27.51 ± 7.92.

In this study we found a high prevalence of AMR genes in the saliva of dental students, supporting the role of the oral cavity as a reservoir for resistance genes. Similarly, Anderson et al. (2023) [28] highlighted the oral microbiome's AMR genes reservoir potential, identifying 64 AMRGs resistant to 36 antibiotics, especially tetracyclines, macrolides, and beta-lactams, through metagenomic sequencing and culture techniques. While Anderson et al. observed greater resistance in healthy and caries-active individuals than in those with periodontitis, our study examined beta-lactamase and aminoglycoside resistance genes specifically within a clinical dental student population, revealing high prevalence but without health-condition distinctions.

Anderson et al. also reported three distinct microbiota ecotypes—two common in healthy and caries-active individuals and one unique to periodontitis—while our study suggests a link between clinical environment exposure and AMR genes prevalence among dental students. Together, these findings highlight the complexity of the oral resistome across health conditions and settings. Anderson et al.'s approach underscores the benefit of multi-method analyses, such as metagenomic sequencing, to capture a broader AMR gene profile—a promising direction for future studies in our population [28].

In a recent relevant study by Bartsch et al. (2024) [29], they explored the impact of chlorhexidine (CHX) mouthwash on oral microbial composition and antimicrobial resistance. In their study, CHX use was linked to a reduction in microbiota diversity and an increase in tetracycline resistance genes, though no significant AMR genes shifts were observed overall. While our study focused on beta-lactamase and aminoglycoside resistance genes in a clinical dental student setting without specific antiseptic use, the findings of Bartsch et al. suggest that exposure to antimicrobials like CHX may selectively influence the oral resistome. This points to the need for further investigation into how antiseptic exposure, common in clinical environments, might affect AMR prevalence among dental students over time, potentially contributing to resistance patterns observed in this population [29].

This study did not find significant associations between participant year group, smoking habits, recent antibiotic usage, and AMR gene prevalence. This suggests that these factors remain stable across academic progression in this sample. Though no statistically significant links were discovered, it is important to keep in mind that this does not rule out the possibility of indirect or subtle relationships that could surface in bigger or more varied populations. Furthermore, even though the study found no appreciable variations in the incidence of AMR

genes, this does not rule out the larger influence of antibiotic use and other behaviors on the emergence and dissemination of AMR genes. Longitudinal studies with a more extensive set of variables, including specific antibiotic classes and precise health metrics, might provide further insights into the dynamics of AMR gene acquisition and prevalence across different demographic or behavioral groups.

The distribution of antimicrobial-resistant genes between first-year and fifth-year dental students was not significantly different. However, a marginal increase in $bla_{CTX-M1}$, $bla_{CTX-M9}$, $bla_{OXA-48}$, $bla_{KPC}$ and $bla_{VIM}$, MOX, $armA$ and $rmtB$ was observed. The genes for ESBL, carbapenemases, AmpC beta lactamases and 16SrRNA methyltransferases are frequently encoded by mobile genetic elements, facilitating their horizontal transfer between bacterial strains and species, thereby contributing to their global dissemination [30]. However, certain disparities in ARG prevalence may be attributed to factors such as geographical location, demographic characteristics, and sampling methods. For example, the relatively low prevalence of $bla_{KPC-1}$ in our study contrasts with reports of higher prevalence in other regions [31], indicating potential regional variations in antimicrobial resistance patterns.

## Conclusion

Results revealed a high occurrence of AMR genes in the oral microbiome, highlighting the complex interplay between genetic determinants of resistance and microbial ecology. A comprehensive metagenomic analysis may be required to evaluate the prevalence and relative amount of antibiotic resistance genes in the oral biofilm samples of the UAE population. This could help establish a database or reference point for these ARGs, enabling effective monitoring in the future.

## Acknowledgments

The dental students in the College of Dental Medicine, University of Sharjah, UAE, for providing saliva samples to conduct this study.

## Author Contributions

**Conceptualization:** Marwan Mansoor Mohammed, Sausan Al Kawas.

**Data curation:** Marwan Mansoor Mohammed, Priyadharshini Sekar, Jahida Al Jamal, Lujayn Abu Taha, Asma Bachir.

**Formal analysis:** Marwan Mansoor Mohammed, Priyadharshini Sekar, Asma Bachir, Sausan Al Kawas.

**Funding acquisition:** Marwan Mansoor Mohammed, Sausan Al Kawas.

**Investigation:** Marwan Mansoor Mohammed, Priyadharshini Sekar.

**Methodology:** Priyadharshini Sekar, Jahida Al Jamal, Lujayn Abu Taha, Asma Bachir.

**Project administration:** Marwan Mansoor Mohammed, Sausan Al Kawas.

**Resources:** Marwan Mansoor Mohammed, Sausan Al Kawas.

**Software:** Asma Bachir.

**Supervision:** Marwan Mansoor Mohammed, Sausan Al Kawas.

**Validation:** Marwan Mansoor Mohammed, Priyadharshini Sekar, Sausan Al Kawas.

**Visualization:** Marwan Mansoor Mohammed, Sausan Al Kawas.

**Writing – original draft:** Marwan Mansoor Mohammed, Priyadharshini Sekar, Jahida Al Jamal, Lujayn Abu Taha, Asma Bachir, Sausan Al Kawas.

**Writing – review & editing:** Marwan Mansoor Mohammed, Priyadharshini Sekar, Jahida Al Jamal, Lujayn Abu Taha, Asma Bachir, Sausan Al Kawas.

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
