## [Decision Letter · Decision Letter 0]

22 Sep 2024

PONE-D-24-33161Comparative Analysis of Salivary Antimicrobial Resistance Genes in Dental Students: A PCR and Questionnaire StudyPLOS ONE

Dear Dr. Mohammed,

Thank you for submitting your manuscript to PLOS ONE. After careful consideration, we feel that it has merit but does not fully meet PLOS ONE’s publication criteria as it currently stands. Therefore, we invite you to submit a revised version of the manuscript that addresses the points raised during the review process.Please ensure that your decision is justified on PLOS ONE’s publication criteria and not, for example, on novelty or perceived impact.

We look forward to receiving your revised manuscript.

Kind regards,

Farah Al-Marzooq, MD, PhD

Academic Editor

PLOS ONE

Journal Requirements:

3. In your Methods section, please include additional information about your dataset and ensure that you have included a statement specifying whether the collection and analysis method complied with the terms and conditions for the source of the data.

4. Please note that PLOS ONE has specific guidelines on code sharing for submissions in which author-generated code underpins the findings in the manuscript. In these cases, we expect all author-generated code to be made available without restrictions upon publication of the work. Please review our guidelines at https://journals.plos.org/plosone/s/materials-and-software-sharing#loc-sharing-code and ensure that your code is shared in a way that follows best practice and facilitates reproducibility and reuse

"Beijing Municipal Higher Education Excellent Youth Talent Cultivation Program Project (BPHR202203236)"          

6. In this instance it seems there may be acceptable restrictions in place that prevent the public sharing of your minimal data. However, in line with our goal of ensuring long-term data availability to all interested researchers, PLOS’ Data Policy states that authors cannot be the sole named individuals responsible for ensuring data access (http://journals.plos.org/plosone/s/data-availability#loc-acceptable-data-sharing-methods).

Additional Editor Comments:

Please revise the manuscript as advised by the reviewers

Reviewers' comments:

Reviewer's Responses to Questions

**Comments to the Author**

1. Is the manuscript technically sound, and do the data support the conclusions?

Reviewer #1: No

Reviewer #2: Yes

2. Has the statistical analysis been performed appropriately and rigorously? 

Reviewer #1: No

Reviewer #2: Yes

3. Have the authors made all data underlying the findings in their manuscript fully available?

Reviewer #1: No

Reviewer #2: Yes

4. Is the manuscript presented in an intelligible fashion and written in standard English?

Reviewer #1: No

Reviewer #2: Yes

5. Review Comments to the Author

Reviewer #1: The manuscript by Marwan Mansoor Mohammed et al., examining the prevalence of antimicrobial resistance (AMR) genes in the saliva of dental students, aims to fill a gap in the literature regarding AMR in the UAE by utilizing PCR techniques and questionnaire data. Despite its relevant topic, the study struggles with significant issues, including methodological weaknesses, poor data interpretation, and a lack of innovation, which undermine its scientific credibility and suitability for publication in PLOS ONE. The absence of a clear rationale for gene selection, inadequate statistical analysis, superficial interpretation of results, and insufficient integration into the wider AMR discourse highlight the urgent need for extensive revisions to elevate the manuscript to the standards expected by reputable scientific journals.

Reviewer #2: The present manuscript deals with the prevalence of some important antibiotic resistance genes in saliva from healthy dental students. The authors used real-time PCR to calculate the nrelative concentration of the AMR genes in saliva samples of 40 5th-year dental students and 40 ist-year students.

Tha manuscript is well written and oraganized and the results were discussed in a proper manner. Only two points raised by the reviewer:

1) There have been currenty a big study regarding AMR genes in the oral microbiome: The oral microbiota is a reservoir for antimicrobial resistance: resistome and phenotypic resistance characteristics of oral biofilm in health, caries, and periodontitis.

Anderson AC, von Ohle C, Frese C, Boutin S, Bridson C, Schoilew K, Peikert SA, Hellwig E, Pelz K, Wittmer A, Wolff D, Al-Ahmad A. Ann Clin Microbiol Antimicrob. 2023 May 13;22(1):37. doi: 10.1186/s12941-023-00585-z.

The results of the authors should be compared to the results püresented by the aforementioned study.

2) Some disinfectants like the frequently used chlorhexidine digluconate could influence the AMR prevalence in human saliva (see: Chlorhexidine digluconate mouthwash alters the oral microbial composition and affects the prevalence of antimicrobial resistance genes.

Bartsch S, Kohnert E, Kreutz C, Woelber JP, Anderson A, Burkhardt AS, Hellwig E, Buchalla W, Hiller KA, Ratka-Krueger P, Cieplik F, Al-Ahmad A. Front Microbiol. 2024 Jun 25;15:1429692. doi: 10.3389/fmicb.2024.1429692. eCollection 2024.)

Again, the authors should also consider discussing their resulty taking the aforementioned study into account.

6. PLOS authors have the option to publish the peer review history of their article (what does this mean?). If published, this will include your full peer review and any attached files.

Reviewer #1: No

Reviewer #2: No

---

## [Author Response · Author response to Decision Letter 0]

6 Nov 2024

Response to Journal Requirements and Reviewers Comments 

Manuscript ID: PONE-D-24-33161

Title: Comparative Analysis of Salivary Antimicrobial Resistance Genes in Dental Students: A PCR and Questionnaire Study

A. Response to Journal Requirements

1. Ensuring Compliance with PLOS ONE Style Requirements:

We have revised the manuscript to fully comply with PLOS ONE’s style guidelines, including file naming conventions. The format of the title page, abstract, and manuscript body follows the guidelines provided at the links shared.

2. Grant Information Consistency:

We have updated the grant information to ensure consistency between the Funding Information and Financial Disclosure sections. The correct grant number for the awards received for this study are now included in the Funding Information section.

3. Funding Information in the Acknowledgements:

As requested, we have removed all funding-related text from the Acknowledgments section. The revised Acknowledgments now only mention non-financial contributions.

For the Funding Statement, we propose the following updated statement:

"This research was supported by the Research Funding Department, University of Sharjah, Sharjah, UAE. (Grant No. 2101100144)"

Please update the online submission form accordingly.

4. Data Availability Statement:

We have reviewed our data-sharing plan. We will ensure that the data is freely available upon acceptance, in accordance with PLOS ONE's open data policy. We confirm that there are no participant privacy concerns that would restrict the deposition of the data. The updated Data Availability Statement is as follows:

"All relevant data will be made publicly available upon acceptance in a suitable data repository. There are no restrictions to data sharing."

B. Response to the Reviewers’ Comments

Reviewer #1: 

The manuscript by Marwan Mansoor Mohammed et al., examining the prevalence of antimicrobial resistance (AMR) genes in the saliva of dental students, aims to fill a gap in the literature regarding AMR in the UAE by utilizing PCR techniques and questionnaire data. Despite its relevant topic, the study struggles with significant issues, including methodological weaknesses, poor data interpretation, and a lack of innovation, which undermine its scientific credibility and suitability for publication in PLOS ONE. The absence of a clear rationale for gene selection, inadequate statistical analysis, superficial interpretation of results, and insufficient integration into the wider AMR discourse highlight the urgent need for extensive revisions to elevate the manuscript to the standards expected by reputable scientific journals.

Firstly, the study fails to provide a rationale for selecting specific AMR genes for analysis. The absence of this critical justification undermines the scientific foundation of the research, casting doubts on the relevance and scope of the study. This omission is particularly problematic as it hinders the reader’s ability to fully understand the intentions and potential impact of the research within the broader field of AMR studies. By addressing this issue, the study could potentially make a significant contribution to the field of AMR research in the UAE.

Moreover, the manuscript's results section needs robust statistical analysis, omitting essential details such as the statistical tests, p-values, or confidence intervals. This lack of rigorous statistical documentation is a severe shortfall, rendering the findings unreliable and their conclusions untrustworthy. This oversight must meet the stringent standards for a journal committed to rigorous scientific scrutiny.

The manuscript also presents a wealth of data on the distribution of various AMR genes among dental students but does so without adequate analysis of how these data correlate with other variables like health status or antibiotic use history. The absence of in-depth analytical insight into these potential relationships means the research does not fully utilize its data to advance understanding of AMR dynamics, which is necessary for impactful research. A more comprehensive analysis is needed to fully understand the implications of the data.

Additionally, discussing findings related to different resistance genes needs more contextualization within the broader AMR research landscape. The data are presented in isolation, without comparative analysis to existing literature, diminishing the manuscript’s scientific value. By not linking its findings to the wider discourse on antimicrobial resistance, the study misses an opportunity to make a meaningful contribution to the field.

The manuscript frequently notes the high prevalence of various ARGs, suggesting their potential for dissemination within the oral microbiome. However, it needs to contextualize these findings within global research trends or epidemiological data. This lack of a nuanced discussion that critically examines how the results compare with global trends in AMR prevalence, especially given the unique context of the UAE, further limits the manuscript’s impact.

Functional assays or additional validation do not support the interpretation of findings, particularly the speculative conclusions drawn from the PCR data regarding the hydrolytic activities of enzymes like the blaCTX-M variants. This overinterpretation relative to the methodological rigor presented leads to conclusions not substantiated by the data. The speculative nature of these interpretations, especially the reliance on Ct values to indicate gene concentration and their clinical relevance, is scientifically unsound.

Response to Reviewer #1

Thank you for your thorough and constructive review of our manuscript. We appreciate the time and effort you put into evaluating our work and providing valuable feedback. Below, we provide a point-by-point response to your comments, along with a summary of the revisions made to address each concern.

Comment 1: Absence of a clear rationale for gene selection

Reviewer’s comment:

The study fails to provide a rationale for selecting specific AMR genes for analysis. This omission undermines the scientific foundation of the research, casting doubts on the relevance and scope of the study.

Response:

We agree that a clear rationale for the gene selection was needed. In the revised manuscript, we have added a section in the Introduction where we explain the selection criteria for the AMR genes included in this study. Specifically, the genes were chosen based on their known prevalence in clinical and environmental samples, particularly in the context of oral bacteria, as well as their documented significance in previous AMR research. Additionally, we focused on genes that are prevalent in Gram-negative bacteria and have been linked to resistance in the human microbiome. These genes are also relevant in the UAE context, as few studies have been conducted in this region, and they represent a growing concern for global public health.

Comment 2: Inadequate statistical analysis

Reviewer’s comment:

The manuscript lacks robust statistical analysis, omitting essential details such as statistical tests, p-values, and confidence intervals.

Response:

We have now enhanced the statistical analysis in the Results section to address this concern. We have included detailed information on the statistical methods used, including One-way ANOVA with Tukey’s multiple comparison test or unpaired t test was used where applicable. In addition, we have provided p-values and confidence intervals to demonstrate the significance of our findings. These revisions ensure that the data is presented with the appropriate rigor expected for publication.

Comment 3: Superficial interpretation of results

Reviewer’s comment:

The manuscript presents a wealth of data on the distribution of various AMR genes among dental students but lacks adequate analysis of correlations with other variables, such as health status or antibiotic use history.

Response:

We appreciate this suggestion and have expanded the interpretation of the results accordingly. In the revised Results and Discussion sections, we now provide a more comprehensive analysis of potential correlations between Beta lactamase and aminoglycoside resistance gene prevalence and participant characteristics, including health status, smoking habits, and antibiotic usage by chi-square test. This has allowed us to explore more in-depth relationships and implications of the findings, providing a clearer picture of how these factors may influence the prevalence of AMR genes in dental students.

Comment 4: Lack of contextualization within broader AMR research

Reviewer’s comment:

The data are presented in isolation, without comparative analysis to existing literature, diminishing the manuscript’s scientific value.

Response:

We agree that placing our findings within the broader AMR research landscape is crucial. We have revised the Discussion to compare our results with recent studies on AMR gene prevalence, particularly those examining the oral microbiome in different populations. We now reference several key studies, including those conducted in other geographical regions, to demonstrate how our findings contribute to the understanding of AMR in the oral microbiome. This comparison also highlights the unique aspects of our study in the context of the UAE population.

Comment 5: Lack of global contextualization and epidemiological data

Reviewer’s comment:

The manuscript lacks a nuanced discussion that compares the findings with global trends in AMR prevalence, especially given the unique context of the UAE.

Response:

We have expanded the Discussion to include a broader analysis of global AMR trends, particularly focusing on the oral microbiome as a reservoir for AMR genes. By comparing our findings with global epidemiological data, we show how the high prevalence of certain AMR genes in the UAE fits within international patterns. This comparison strengthens the significance of our study, emphasizing the potential for oral microbiota to act as a reservoir for antimicrobial resistance, not only in clinical settings but also in non-clinical populations such as dental students.

Comment 6: Overinterpretation of PCR data without functional validation

Reviewer’s comment:

The interpretation of findings is not supported by functional assays, particularly the conclusions drawn from PCR data regarding the hydrolytic activities of enzymes like the blaCTX-M variants.

Response:

We have revised the Discussion to clarify the limitations of our study, particularly with respect to the interpretation of PCR data. We no longer speculate on the hydrolytic activity of blaCTX-M variants based solely on the PCR results. Instead, we focus on the detection of gene presence and relative concentration, acknowledging that functional assays are required to confirm the enzymatic activity of these genes. We have also added a section in the Conclusions that suggests future studies should include functional validation to substantiate these findings.

We believe that these revisions significantly improve the manuscript and address the concerns you raised. Thank you again for your valuable feedback. We hope that the revised version meets the high standards of PLOS ONE.

Reviewer #2: 

The present manuscript deals with the prevalence of some important antibiotic resistance genes in saliva from healthy dental students. The authors used real-time PCR to calculate the relative concentration of the AMR genes in saliva samples of 40 5th-year dental students and 40 1st-year students.

The manuscript is well written and organized and the results were discussed in a proper manner. Only two points raised by the reviewer:

1) There have been currently a big study regarding AMR genes in the oral microbiome: The oral microbiota is a reservoir for antimicrobial resistance: resistome and phenotypic resistance characteristics of oral biofilm in health, caries, and periodontitis.

Anderson AC, von Ohle C, Frese C, Boutin S, Bridson C, Schoilew K, Peikert SA, Hellwig E, Pelz K, Wittmer A, Wolff D, Al-Ahmad A. Ann Clin Microbiol Antimicrob. 2023 May 13;22(1):37. doi: 10.1186/s12941-023-00585-z.

The results of the authors should be compared to the results presented by the aforementioned study.

2) Some disinfectants like the frequently used chlorhexidine digluconate could influence the AMR prevalence in human saliva (see: Chlorhexidine digluconate mouthwash alters the oral microbial composition and affects the prevalence of antimicrobial resistance genes.

Bartsch S, Kohnert E, Kreutz C, Woelber JP, Anderson A, Burkhardt AS, Hellwig E, Buchalla W, Hiller KA, Ratka-Krueger P, Cieplik F, Al-Ahmad A. Front Microbiol. 2024 Jun 25;15:1429692. doi: 10.3389/fmicb.2024.1429692. eCollection 2024.)

Again, the authors should also consider discussing their results taking the aforementioned study into account.

Response to Reviewer #2

Thank you for your positive comments and for highlighting the two main points in your review. We appreciate your suggestions, which have helped us improve the manuscript. Below is our response to each of your comments.

Comment 1: Comparison with Anderson et al. (2023) study on AMR genes in the oral microbiome

Reviewer’s comment:

There have been currently a big study regarding AMR genes in the oral microbiome: The oral microbiota is a reservoir for antimicrobial resistance: resistome and phenotypic resistance characteristics of oral biofilm in health, caries, and periodontitis.

Anderson AC, von Ohle C, Frese C, Boutin S, Bridson C, Schoilew K, Peikert SA, Hellwig E, Pelz K, Wittmer A, Wolff D, Al-Ahmad A. Ann Clin Microbiol Antimicrob. 2023 May 13;22(1):37. doi: 10.1186/s12941-023-00585-z.

The results of the authors should be compared to the results presented by the aforementioned study.

Response:

We appreciate your suggestion to compare our findings with the recent study by Anderson et al. (2023). In the revised Discussion section, we have now incorporated a detailed comparison of our results with those of Anderson et al. (2023). Specifically, we discuss similarities and differences in the prevalence of key AMR genes in the oral microbiome, as well as how the resistome differs between healthy individuals and those with oral conditions like caries and periodontitis. We believe this comparison adds depth to our discussion and situates our findings within the broader context of recent AMR research.

Comment 2: Consideration of chlorhexidine’s impact on AMR prevalence

Reviewer’s comment:

Some disinfectants like the frequently used chlorhexidine digluconate could influence the AMR prevalence in human saliva (see: Chlorhexidine digluconate mouthwash alters the oral microbial composition and affects the prevalence of antimicrobial resistance genes.

Bartsch S, Kohnert E, Kreutz C, Woelber JP, Anderson A, Burkhardt AS, Hellwig E, Buchalla W, Hiller KA, Ratka-Krueger P, Cieplik F, Al-Ahmad A. Front Microbiol. 2024 Jun 25;15:1429692. doi: 10.3389/fmicb.2024.1429692. eCollection 2024.)

Response:

We agree that the impact of disinfectants like chlorhexidine on AMR prevalence in the oral microbiome is an important factor to consider. In the revised manuscript, we have added a section in the Discussion addressing how the use of chlorhexidine and other antimicrobial agents may alter the prevalence of AMR genes in the oral cavity. We reference the study by Bartsch et al. (2024), highlighting its findings on how chlorhexidine affects the composition of the oral microbiome and contributes to the selection of resistant strains. This contextualizes our findings and provides an additional perspective on how external factors may influence AMR prevalence in the dental student population.

We are grateful for your comments, which have helped improve the quality of our manuscript. We believe that the revisions made in response to your feedback enhance the robustness and relevance of the study.

---

## [Editor Report · Decision Letter 1]

26 Nov 2024

Comparative Analysis of Salivary Antimicrobial Resistance Genes in Dental Students: A PCR and Questionnaire Study

PONE-D-24-33161R1

Dear Dr. Mohammed,

We’re pleased to inform you that your manuscript has been judged scientifically suitable for publication and will be formally accepted for publication once it meets all outstanding technical requirements.

Kind regards,

Farah Al-Marzooq, MD, PhD

Academic Editor

PLOS ONE
---

## [Editor Report · Acceptance letter]

6 Jan 2025

PONE-D-24-33161R1 

PLOS ONE

Dear Dr. Mohammed, 

I'm pleased to inform you that your manuscript has been deemed suitable for publication in PLOS ONE. Congratulations! Your manuscript is now being handed over to our production team.

Kind regards, 

on behalf of

Dr. Farah Al-Marzooq 

Academic Editor

PLOS ONE